# Application of Cold Storage and Short In Vitro Germination for Somatic Embryos of *Pinus radiata* and *P. sylvestris*

**DOI:** 10.3390/plants12112095

**Published:** 2023-05-24

**Authors:** Cathie Reeves, Mikko Tikkinen, Tuija Aronen, Jana Krajnakova

**Affiliations:** 1Scion, Te Papa Tipu Innovation Park, 49 Sala Street, Private Bag 3020, Rotorua 3046, New Zealand; jana.krajnakova@scionresearch.com; 2Natural Resources Institute Finland (Luke), FI-57200 Savonlinna, Finland; mikko.tikkinen@luke.fi (M.T.); tuija.aronen@luke.fi (T.A.)

**Keywords:** somatic embryogenesis, *Pinus radiata*, *Pinus sylvestris*, germination, acclimatization, cold storage, embling quality

## Abstract

Somatic embryogenesis (SE) is an advanced vegetative propagation technology that, when used in combination with breeding and cryopreservation, offers the forest industry a powerful tool for the deployment of elite genotypes. Germination and acclimatization are critical and cost-intensive phases in the production of somatic plants. The efficient conversion of somatic embryos into robust plants is a necessity if a propagation protocol is to be successfully adopted by the industry. In this work, these late phases of the SE protocol of two pine species were investigated. A shortened germination protocol and more controlled acclimatization were investigated for Pinus radiata, testing embryos from 18 embryogenic cell lines. A more simplified protocol, including a cold storage phase, was also compared among 10 of these cell lines. A shortened germination period and more controlled protocols significantly improved the acclimatization of somatic embryos directly from the lab to the glasshouse. When results for all cell lines were pooled, there were significant improvements in all growth characteristics (shoot height, root length, root collar diameter, and root quadrant score). When the more simplified protocol involving cold storage was tested, improvements were seen in the root architecture. For *Pinus sylvestris,* the late phases of somatic embryogenesis were investigated on seven cell lines in a set of two trials (four to seven cell lines per trial). During the germination phase, a shortened and simplified in vitro period, a cold storage option and basal media were explored. Viable plants were obtained from all treatments. However, there is still the need to improve germination and related protocols together with growing regimes for *Pinus sylvestris*. The improvements to protocols presented here, particularly for *Pinus radiata*, result in greater survival and quality of somatic emblings, leading to reduced costs and increased confidence in the technology. Simplified protocols using a cold storage option show great promise and, with some further research, could lead to reductions in the cost of the technology.

## 1. Introduction

Radiata pine (*Pinus radiata* D. Don), a native of North America, is the most widely planted pine in the world. It is extensively cultivated as a plantation timber in many temperate parts of the world, including Australia, Spain, Argentina, Chile, Uruguay, Kenya, South Africa, and New Zealand (NZ), where it is the main commercial forestry species, occupying approximately 87% of plantation forests. In NZ, most tree stocks produced (64%) originate from control pollinated seed, including seedlings, cuttings, and clones. The remaining (36%) originate from OP seed [1]. In vitro propagation protocols for radiata pine have been developed in New Zealand in order to increase the availability of material from elite crosses, firstly via adventitious shoots from mature seeds [2] and, more recently, somatic embryogenesis (SE), from immature seeds [3].

Scots pine (*Pinus sylvestris* L.) is the most common species of the genus *Pinus* and an economically important species in the Northern Hemisphere. As wood, it is used for building, construction, and furniture, with further processing as veneer, fiberboards, or chipboards, and for pulp and paper production. Its natural range is broad, covering large areas with different climatic conditions in Eurasia, from the West Coast of Europe to the far east of Russia. Within its range, Scots pine is often the dominant species and can occupy up to 65% of the forest area. Within the EU, nearly all the forest reproductive material comes from approved basic materials and is either improved seed orchard seed or seed from selected seed stands. For forest regeneration, both planting and direct seeding are used [4]. Scots pine is considered a recalcitrant propagator via cuttings, but SE from immature seed has proved successful and is currently used for research purposes [5].

SE of conifers is a multistep process, including the initiation of embryogenic masses, the proliferation of these masses, the formation of mature somatic embryos (SEs), germination of somatic embryos, and finally, acclimatization to ex vitro conditions in preparation for field planting. A large amount of work has been done over the last few decades to improve SE protocols in conifers, with most advances occurring in *Picea* species due to its relative ease of propagation. *Pinus* species have been more recalcitrant to SE [6]. Despite this, improvements are continuously being made as protocols are refined [7].

When it comes to improving the quality of somatic emblings in conifers, most published studies have focused on the maturation stage of the SE process. The maturation process in culture is initiated with the development of the somatic embryos and terminates with an imposed desiccation prior to germination. For complete maturation to occur, the embryos must achieve both “morphological” and “physiological” maturity [8,9,10]. The regeneration of morphologically developed somatic embryos does not guarantee satisfactory post-embryogenic performance unless defined physiological conditions are attained. Factors during maturation that have been shown to have an influence on the success of germination include (i) nutritional factors such as type of carbon and nitrogen sources; (ii) maturation media components that affect gel strength; (iii) abscisic acid (ABA) concentration and type; (iv) type and timing of osmotic agents, and (v) type and time in contact with plant growth regulators [11,12,13].

The stages of germination and conversion of emblings into plants are extremely important to the overall success of SE on a commercial scale. Loss of plants during these latter stages greatly increases the overall cost of the technology, with the germination step being the most costly due to high labor demands [14,15,16]. Despite this, until recently, these stages of conifer SE protocols (germination and early plant growth) have been largely neglected in research efforts [8]. Although the germination and acclimatization stages are often reported to a lesser degree in many studies [9], where more detailed descriptions are given, germination protocols are often lengthy [17,18,19], with some containing multiple transfers of embryos onto fresh germination medium [18,20,21], and protocols are often tested in only a small number of cell lines [22,23]. It is not until recently that more detailed studies regarding the germination stage of SE in conifers have started to emerge [24,25]. Successful germination and acclimatization have been shown to be highly influenced by several factors during these latter two stages of the process. These include (i) effect of genotype; (ii) source of somatic embryos (age of embryogenic cell line); (iii) embryo morphology [26]; (iv) nitrogen source [24,25,27]; (v) light quality and quantity [24,25,28,29]; (vi) cold storage [24]; and (vii) time on germination media [24].

In the current *P. radiata* SE process, the resilience of tissue-cultured plants, when transferred to the nursery, is poor. Root systems developed are often a limiting factor for their clonal deployment potential due to scarcity of roots, cessation of their growth and unbalanced architecture [21,30]. In NZ, to circumvent this problem germinated somatic embryos are often not used directly and are instead put through an extra in vitro multiplication step prior to acclimatization, allowing the formation of a more robust micro-shoot culture, more capable of successful acclimatization [7]. However, this extra step adds undesired time and cost to the overall process and automation of this process would be extremely difficult. Likewise, SE-plants of *P. sylvestris* regenerated following a long, multistep in vitro germination process has been shown to have unbalanced root architecture that may result in bent stems and leaning trees, although suboptimal nursery techniques may have added to these problems [5].

Within the last five years, a mounting number of studies in conifers looking at improving germination and acclimatization stages of the SE process have been published. Of particular interest are detailed studies of physical and chemical growing conditions during the germination process in *Picea abies* (Norway spruce) SE [16,24]. In this work, the authors showed that cold storage of good quality, mature somatic embryos, followed by a very short in vitro germination period, significantly improved the survival and continued growth of Norway spruce somatic emblings. The best treatment combination of cold storage on filter paper, lower nitrogen content in the germination medium and one-week in vitro germination resulted in an 88% higher survival and 28% higher growth compared to the poorest reference treatment in the same test year. These emblings could be planted after a nursery period one year sooner than that of the control emblings. The results indicate that Norway spruce emblings germinated for one week in vitro can be transplanted and grown in nurseries without any additional treatments or environmental control differing from seedlings, which is a prerequisite to reach standards for forest regeneration material.

The objective of the work presented here was to test a shortened germination protocol developed for Norway spruce [24] as well as the use of micro-plugs (Vivi™ trays, Burgh Haamstede, The Netherlands) for the acclimatization of *P. radiata* and *P. sylvestris* SE plantlets. A much-simplified protocol was also investigated for both species, where somatic embryos were placed in cold storage prior to short germination on Petri plates and then direct transfer to nursery conditions. The effect of these protocols on the early stages of growth and survival of *P. sylvestris* and on survival, growth characteristics and root architecture of four-to-five-month-old somatic emblings of *P. radiata* is reported.

## 2. Results

### 2.1. Pinus radiata

At the end of maturation (10–12 weeks), cotyledonary somatic embryos from 18 cell lines were put through two germination/acclimatization treatments: control (a five-week in vitro germination in Petri plates followed by transfer to propagation trays in a glasshouse) and a short germination/micro-plug treatment (Trt 1, a two week in vitro germination in Petri plates followed by transfer to a micro-plug system under sterile conditions for three weeks before being transferred to a glasshouse). At the same time, somatic embryos from 10 of these cell lines were also stored in cold storage, followed by a short germination (two weeks on Petri plates) and planting directly into nursery treatment (Trt 2). All treatments are described by illustration in Figure 1 below. The number of somatic embryos varied per cell line and treatment depending on availability, ranging from 10 to 72 embryos (Appendix A).

#### 2.1.1. Survival of Somatic Emblings in Different Germination/Acclimatization Protocols

Survival at the repotting stage is indicative of successful germination/conversion (formation of a root and epicotyl growth) and early acclimatization. A combination of a reduced time somatic embryos spent in a germination medium and the inclusion of a micro-plug system had a significantly positive effect on the survival (*p* < 0.001) of somatic emblings by the repotting stage. When all cell line data were combined, 86% of emblings germinated using this protocol had survived and were able to be repotted, compared to only 64% in the control protocol (Figure 2). Trt 2 (cold storage/short germination protocol) did not have a repotting stage, with germinated embryos being planted directly in the final container. Individual cell line survival rates at the repotting stage ranged from 23–100% in the control treatment and 50–100% in Treatment 1 (short germination/micro-plug protocol) (Appendix A).

Survival by the end of the experiment is indicative of successful acclimatization from the propagation tray environment (Control) and micro-plug environment (Trt 1) and successful conversion and acclimatization in Trt 2 (cold storage/short germination protocol). The survival of emblings by the end of the experiment germinated and acclimatized under the three treatments differed significantly (*p* < 0.0001). The highest survival was 85% of emblings grown in Trt 1 (short germination/micro-plug protocol), followed by those grown in the control treatment and then those grown in Trt 2 (cold storage/short germination protocol), at 59% and 38%, respectively (Figure 3).

Individual cell line final survival percentages ranged from 9–100% in the control treatment, 41–100% in Trt 1, where all but one cell line ranged from 75–100%, and 5–73% in Trt 2 (Appendix A, Figure 4).

#### 2.1.2. Growth Characteristics of Surviving Somatic Emblings

Cell line and germination treatment significantly affected the shoot height, root length and root collar diameter (rcd) of *P. radiata* somatic emblings by the end of the experiment (Table 1).

When cell line data were pooled, emblings that were germinated and acclimatized using the short germination/micro-plug protocol (Trt 1) had significantly taller shoots (78.37 mm), longer roots (126.58 mm) and higher root collar diameters (1.28 mm) on average than emblings from both the control (47.72 mm, 110.08 mm and 1.17 mm, respectively) and the cold storage/short germination protocol (Trt 2) (51.91 mm, 112.27 mm, and 1.71 mm, respectively) (Figure 5A–C, respectively).

Shoot height was also significantly affected by cell line, with 15 out of the 18 cell lines tested in the control protocol having significantly shorter emblings on average than those from Trt 1 (short germination/micro-plug protocol). The other three cell lines (N, O and R) showed no significant differences in average shoot height between the two treatments. When the average shoot heights of the 10 cell lines which went through all 3 treatments were compared, Trt 2 (cold storage/short germination protocol) resulted in shoots that were not significantly taller than those from the control treatment for 7 of these lines. The other three lines resulted in shoots that were not significantly taller than those from Trt 1 (short germination/micro-plug protocol) (Appendix A).

For root length, 8 out of the 18 cell lines had significantly longer roots when germinated and acclimatized using Trt 1 compared to those from the control. The other 10 cell lines, although not significantly different, still followed that same trend. Of the 10 cell lines that went through all 3 treatments, 4 showed no significant difference between the three treatments. For the remaining six cell lines, emblings germinated and acclimatized using Trt 2 had significantly shorter roots than those from Trt 1 (Appendix A).

Average root collar diameters were significantly larger for 7 out of the 18 cell lines which were germinated and acclimatized using Trt 1 compared to those from the control treatment. The remaining 11 cell lines showed no significant difference between the two treatments, although the same trend was seen. For the 10 cell lines which were also included in Trt 2, 5 showed no significant differences in average root collar diameters in emblings from the 3 treatments, and 4 were significantly lower than emblings from Trt 1 (Appendix A).

#### 2.1.3. Balance of Root System (Root Quadrant Score) of Surviving Somatic Emblings

Root quadrant scores from all surviving emblings from each of the germination/acclimatization protocols were separated into two groups representing either poor root architecture (a score of 0, 1 or 2) or good architecture (a score of 3 or 4). When embryos were germinated and acclimatized under Treatment 2 (cold storage/short germination protocol), 72% of emblings fell into the good root architecture category, compared to only 15% of emblings from Treatment 1 (short germination/micro-plug protocol) and 10% of emblings from the control treatment (Figure 6 and Figure 7).

### 2.2. Pinus sylvestris

Maturation was carried out for seven cell lines in October and in November 2020. From the maturation carried out in October, 216 cotyledonary embryos from four cell lines (54 embryos per cell line) were germinated in two treatments directly following maturation in December 2020 (DEC20 trial). The DEC20 trial emblings were transplanted to micro-plugs following germination. All remaining embryos (870 embryos from seven cell lines from both maturation events) were cold stored until June 2021, germinated in three germination treatments and transplanted to nursery conditions in 85 cm^3^ container plugs (June21 trial). All treatments are described in the illustration below in Figure 8.

#### 2.2.1. Embryo Size and Growth during Cold Storage and Germination

The cold stored cotyledonary embryos (June21 trial) were significantly longer (*p* < 0.01) than the non-cold stored cotyledonary embryos (Dec20 trial), at 2.38 ± 0.02 mm and 2.18 ± 0.04 mm, respectively (Figure 7). The size of the embryos varied significantly among cell lines in both trials (*p* < 0.01, in all, Figure 9).

After one week of in vitro germination (DEC20 trial), Scots pine emblings were found to be approximately half of the length of Norway spruce emblings (seen in previous work) after the same duration of in vitro germination. This supported the initial plan to continue the in vitro germination for the second week. For the Dec20 and June21 emblings, when cell line data were pooled, mean shoot lengths after two weeks of in vitro germination were 3.20 ± 0.09 mm and 2.80 ± 0.05 mm (*p* < 0.01), respectively (Figure 10). Mean root lengths of the Dec20 and June21 emblings after a two-week in vitro germination were 2.14 ± 0.12 mm and 2.46 ± 0.10 mm (*p* = 0.81), respectively (Figure 10). The mean total length of emblings in the two germination trials, Dec20 and June21, after a two-week in vitro germination were 5.34 ± 0.20 mm and 5.26 ± 0.13 mm (*p* < 0.01), respectively (Figure 10).

In the December 2020 trial (Dec20), the emblings varied in size among cell lines in shoot length (*p* < 0.01), root length (*p* = 0.014) and total length (*p* < 0.01, Appendix A). No difference in the embling size between treatments was observed after two weeks of in vitro germination (Appendix A). In the June 2021 trial (June21), the embling root length varied among treatments after two weeks of in vitro germination (*p* < 0.01, Appendix A). The roots were significantly longer in LM 2 w and in the Control treatment than in MB 2 w (*p* < 0.01, in both, Appendix A).

All embryos were transplanted in both trials to growing containers (micro-plugs or Plantek Pl 81 s). Thus, embryos that could have been interpreted as not germinated were also transplanted to peat to enable tracking of the development of individual embryos. We attempted to apply a threshold in size during in vitro germination to define if the embryo has achieved germination or not by setting the following size limits: Overall length minimum of 4 mm, of which root must be at least 2 mm (they had roughly doubled their size and had visible root growth) (Table 2).

#### 2.2.2. Survival of Somatic Emblings in Different Germination/Acclimatization Protocols

Post-transplant survival of emblings in the DEC20 trial was 46.8% overall (Table 2). In the Control and MB 2 w treatments, survival rates were 48.1% and 45.4%, respectively (Table 2). No statistical difference in survival was found between treatments. The length of the embling shoot, root and overall, after two weeks of germination, had a significant effect on post-transplant survival (*p* ≤ 0.01 in all, Figure 11).

In the Dec20 trial, when the emblings were grouped in size classes, survival was significantly lower when the embling length was below 3 mm than in all other groups (*p* < 0.03 in all, Figure 11c). Also, when embling length was 3–6 mm, survival was significantly lower than among groups of longer emblings (*p* ≤ 0.02 in all). When root lengths were grouped (2 mm length groups), the survival was lower when root length was below 2 mm than when root length was 2–6 mm (*p* ≤ 0.01, in all, Figure 11b). Additionally, the survival in the group where the root length was 2–4 mm was lower than when the root length was 4–6 mm (*p* = 0.01). When the shoot length was between 2–8 mm, the survival was significantly higher than when the shoot length was 1–2 mm (six groups, (*p* < 0.02 in all), Figure 11a. Survival was also higher when the shoot length was 4–8 mm (four groups) than when the shoot length was less than 1 mm (*p* ≤ 0.02 in all). Additionally, the survival was significantly higher when the shoot length was 6–7 mm (three groups) than when the shoot length was 3–5 mm (*p* < 0.05, in all).

In the June21 trial, post-transplant survival was low overall (15.3%, Figure 11c). Survival was significantly higher in the control treatment than in other treatments (*p* < 0.01 in both, Table 2). The length of embling shoot, root and overall, after two weeks of germination, had a significant effect on post-transplant survival (*p* < 0.01 in all, Figure 11c). In the June21 trial, when the total embling length was under 9 mm, the survival was lower than among longer emblings (*p* ≤ 0.05, in all except when the length was from 18 to less than 24 mm). When the root length was less than 2 mm, survival was significantly lower than when the root length was 4–6 mm (*p* = 0.01, Figure 11b). Also, the difference in survival was significant when root length was 8–10 mm, 18–20 mm or 24–26 mm than when root length was less than 4 mm, or 6–8 mm (*p* ≤ 0.05 in all, Figure 11b). A significant difference between groups with root lengths of 24–26 mm and when root length was more than 26 mm was found (*p* < 0.05). In shoot length groups, the survival was significantly higher when shoot length was from 6–8 mm or more than 11 mm than in other groups (*p* < 0.05, in pairwise comparisons, Figure 11a). Also, higher survival was found when shoot length was 5–6 mm than when shoot length was 4–5 mm or less than 1 mm (*p* < 0.05, in both comparisons). Additionally, survival was higher when shoot length was 2–4 mm than when shoot length was under 1 mm (*p* < 0.05 in both comparisons).

## 3. Discussion

Large-scale uptake of SE into breeding programs of commercially grown conifers is currently being hindered by a number of factors. These factors can be condensed into two overarching areas of concern for the grower; the cost and the quality of the final product [31,32]. Although much research on SE protocols of economically important forestry species has been carried out, when it comes to the cost of the process, much less is known [15]. In a review of the benefits and risks of clonal forestry, Wu [33] presented the costs of tested clones of radiata pine in NZ and loblolly pine in the US, showing them to be significantly more costly than CP seedlings in both species. There have been a very limited number of more comprehensive assessments of the costs involved with SE. In studies that have attempted to break the costs down, Cervelli and Senaratna [34] and Tikkinen [16] both concluded that the latter stages in the SE process of spruce species (desiccation, germination, and conversion) were the most costly due to the current need for handling individual plantlets during these stages. The extra cost for SE material reflects the high costs of the current technology but is also (where field tests have been carried out) reflective of the nature and amount of genetic improvement seen with clones [35,36]. In a survey carried out with Finnish forest owners and professionals, Tikkinen et al., showed that there was a willingness to pay for improved traits [37]. Despite this, there is still the belief that in order for SE to be commercially feasible, the cost of a somatic embling needs to be close to the cost of a seed orchard seedling [15]. Although this goal seems rather unlikely currently, with rapid advancements in automation technology occurring [38], it may become more likely.

SE of *P. radiata* in New Zealand is currently being used on a small scale by some of the larger forestry companies. Hesitation around more widespread uptake is mostly based on the cost of the final product. Propagation of *P. radiata* rooted cuttings from CP material is well established in NZ, and most forestry growers are happy to pay a premium price for them to receive a sturdier plantlet, especially for harsher or more fertile sites [30]. Incorporation of the rooted cuttings technology into the SE process was, therefore, a logical progression and has been presented as a cost-effective delivery system [35]. This propagation strategy is also being used in Ireland for Sitka spruce [15]. However, current protocols used by industry in NZ often also involve an SE-organogenesis hybrid method to produce the mother plants for stool beds [7]. While this strategy gets around historical issues seen with somatic embling quality [30], it also adds to the overall cost of the final product and will likely limit the possibilities of integration into automated systems. It also means that there have been a very limited number of studies done on improving germination and acclimatization using direct somatic embryos. Recently Castander-Olarieta and co-workers carried out some maturation and germination studies with radiata pine direct somatic embryos. They looked at the effect of some medium modifications at the proliferation, maturation and germination stages and their effects on plant conversion and survival [25]. However, this study did not look closely at the quality of the resulting plants following acclimatization. If high-quality somatic emblings are capable of either being used as a mother plant for cutting production or, preferably, directly transferred to the field, avoiding the need for further multiplication where possible, this would obviously be the cheapest [39] and fastest option.

The aim of the work presented here was to test a shortened germination protocol developed for Norway spruce [24] on two pine species, *P. radiata* and *P. sylvestris*, as well as to improve the acclimatization protocol and the overall quality of somatic emblings of *P. radiata*. A much-simplified protocol was also attempted for both species, where somatic embryos were placed in cold storage prior to short germination on Petri plates and then directly transferred to nursery conditions. For Scots pine, embryos were measured at the end of maturation (with and without cold storage) as well as after germination and survival, with data recorded after the first growing season. For radiata pine emblings, a morphological assessment was carried out, including some of the standard measurements made to assess the quality of seedlings [40], shoot height, root length, root collar diameter, and root architecture of four-month-old emblings in the glasshouse.

A short in vitro germination on Petri plates, coupled with the use of a micro-plug system designed to assist with the control of humidity, water availability, and gas exchange in the substrate while growing in vitro plantlets (http://www.vivi.nu, accessed on 23 March 2023), has been successfully applied to both radiata pine and Scots pine. The majority of published SE germination protocols in pines are lengthy. *P. taeda* somatic embryos are cultured on a germination medium for 7 to 8 weeks in total [17], up to 12 weeks for Japanese pines [19] and for 14 to 16 weeks for *P. pinaster* [18]. In this work, we successfully reduced the period from the normal four to five weeks to just two weeks on germination media before planting emblings into sterile soil in micro-plugs in both tested species. The majority of embryos germinated (roots had emerged) during this two-week period, with conversion (both root and epicotyl emergence) continuing in the micro-plug environment.

In this work, we are unable to separate the effect of cold storage on somatic embryo germination and acclimatization from other factors (short germination/use of micro-plugs) due to the vastly different propagation protocols used in the treatments tested. Despite this, cold storage has been shown to have no adverse effects on germination in some conifers [25,41] and may, in fact, have some positive effects on preparing the embryos for germination and acclimatization, such as reduction of endogenous ABA [42] and accumulation of storage proteins [43,44]. Cold storage is also a commonly-used strategy for allowing flexibility for embling production when large numbers are required in a small window of time [39]. Unfortunately, no measurements were taken on radiata pine embryos before and after cold storage. From the measurements taken on Scots pine somatic embryos following maturation (before cold storage) and again after six to seven months of cold storage (Figure 9), we saw that the mean length of cotyledonary embryos was significantly (*p* < 0.01) higher following cold storage (2.18 and 2.38 mm, respectively), suggesting some continued development of embryos. Although the size increment is small, it could be an indicator of changes in storage compounds or even precocious germination [44]. The latter would indicate that the cold storage conditions were not ideal or may have been too long in this work. The observed lower survival in the control treatment on emblings in the June21 trial may partially be due to suboptimal cold storage but could also be due to the environmental conditions following the transplant, which were much harsher (direct transplant to a commercial nursery in 85 cm^3^ containers) than in the DEC20 trial, where they were planted into the controlled micro-plug environment. Although no measurements were made of radiata pine embryos before and after cold storage in this work, it was clear to see that some development of embryos continued during the five to six months in cold storage. Optimum cold storage conditions and timing should be investigated more closely for these species. Varis et al., reported no significant differences in germination rates between somatic embryos that germinated immediately following maturation and those cold stored for three to six months and even up to twelve months (although some precocious germination was seen when embryos were stored for this long) in Norway spruce [41]. Recently, Välimäki et al., suggested an optimal cold storage time of up to eight weeks for Norway spruce SE, with longer periods not being detrimental if required [44]. Castander-Olarieta et al., found no significant differences in the dry weight of radiata pine somatic embryos before and after cold storage of eight weeks and suggested that it is possible to store them for up to nine months. However, no data was presented to confirm this [25].

Applying a germination medium refined for Norway spruce (nitrogen content lowered and the source being purely organic) appears to support root growth in Scots pine. However, more studies are required to confirm this. Perhaps similar protocol modifications would be favorable for the media conventionally applied in Scots pine SE. Such a reduction or removal of inorganic nitrogen from germination media has been found to be favorable in several conifer species [9,24,25,27,45].

For radiata pine, a shortened germination period and the use of a micro-plug system to assist with acclimatization significantly (*p* < 0.001) improved survival of somatic emblings compared to the control treatment at the time of repotting, 86 and 64%, respectively (Figure 2). Surviving emblings grown in either propagation trays (control) or micro-plugs (Trt 1) were repotted into TS48 trays and monitored for a further 13–15 weeks. The effect of treatment on survival continued to be seen in some cell lines during this second stage of propagation, where for half of the cell lines tested, most emblings were lost (especially when grown under control conditions, Appendix A). These losses were likely to be due to emblings from the control treatment not being as well adapted to the ex vitro environment [46] or containing poorly developed root systems and therefore suffering from transplant shock [47]. It is also possible that the transplanting procedure itself could result in plant loss, although this type of loss is more likely when the quality of the plant is less than optimal [48]. When a more simplified protocol (Trt 2) was applied to radiata pine, the final survival percentage of somatic emblings was only 38% (Figure 3). Although this was lower than the embling survival seen in both the control and Treatment 1 by the end of the experiment, 59 and 85%, respectively, it was similar to the control treatment when differences in the calculation of final survival percentages were considered. This result was not unexpected due to the harshness of Treatment 2 (direct transfer of 2-week-old emblings to the nursery) compared to the other two treatments.

For Scots pine, survival results varied (Table 2), suggesting that protocol improvements in earlier phases of the SE process are still required for this species. This is also clear from previous work, where embryo quality has been shown to vary and affect the germination and acclimatization outcome [25,41]. The size of the somatic embryo prior to and following two weeks of germination was measured, and survival was followed for the first growing season. The highest transplant survival rates were obtained when the overall length of emblings was at least 3 mm or when at least 2 mm of the root was seen following two weeks of germination. When emblings were transplanted into miniplugs in DEC20, the emblings from the control (two-step in vitro method) and the ones from shortened in vitro germination survived equally well in the greenhouse (48 and 45%, respectively). Previously, survival varying from 10 to 90% has been observed using the present control treatment, depending on the quality of the embryos selected for germination [12]. Recently, using a method similar to our control, Harju et al., reported an in vitro germination rate of 32% for open-pollinated Scots pine SE material and 53% for lines from controlled crossings, i.e., within the range of our controls, at 37 to 41% germination [49]. For Harju et al., material, overall embling greenhouse survival was 74% of germinated embryos. If a similar calculation was applied in DEC20 using a length threshold of 3 mm overall length with at least 2 mm of root length for successful germination, the mean survival was 66.7%, 57.5% and 78.1% in control and MB2 w treatments, respectively. When a more simplified protocol was applied to Scots pine (June21 trial), survival after the first growing season was greatly reduced, at 10.2% and 7% for the LM2 w and MB2 w treatments, respectively. The survival rate of emblings from the control treatment (cold storage, two-step in vitro germination, and planting directly into commercial containers in the nursery) was 28.6%. However, it must be considered that all embryos which were placed into germination were transplanted to peat regardless of whether they had germinated or not. If a length threshold of 3 mm overall, with at least 2 mm of the root, is applied to determine germination in the present study, the mean survival was 22.3%, 29.7% in control, and 16.3% and 17.9% in LM2 w and MB2 w treatments, respectively, while the overall germination percent was 32.9, being nearly equal to the germination percentage in DEC20. The results indicate that short plate germination improved survival in Dec20 emblings, and in June21 emblings, higher survival was obtained from a two-step in vitro germination. One possible explanation for this may be the container plug size, which was much larger in the June21 trial (85 cm^3^) than in the DEC20 trial (3.4 mL). The July conditions in a commercial nursery in Finland are very harsh and have certainly decreased the overall survival. The results from measurements taken following germination on plates for two weeks suggest that when planting directly to larger containers (85 cm^3^), the emblings should be larger, the roots should be at least 6 mm long, or the total length of emblings should be at least 9 mm to increase the chances of survival. This finding is supported by the result that emblings from the two-step germination survived better in the June21 trial, likely due to their larger size, i.e., longer germination, even though they were the same size after the plate germination. The argument regarding suitable embling size for certain container plug sizes is supported by the survival results from the Dec20 trial, where no additional benefit was achieved by using the two-step in vitro germination. Although the results must be verified with a larger sample size, these initial results could then be applied to improve the germination protocols and transplanting success into different containers and perhaps applied to other species, such selection criteria could help with the integration of SE laboratory protocols with greenhouse protocols [32].

Applying a propagation protocol to radiata pine involving short germination on Petri plates, transfer of emblings to a micro-plug system to assist with acclimatization and transplanting surviving emblings into TS48 trays in nursery conditions (Trt 1) had a significant effect (*p* < 0.0001) on all growth characteristics measured (shoot height, root length and root collar diameter) (Table 1, Figure 5). Although there were some differences in how cell lines behaved, even when there were no statistical significances seen, the same trend was often observed in these lines (Appendix A). It is important that somatic emblings be of a suitable quality to be able to be taken through nursery protocols to produce a finished plant that meets all the normal quality standards applied to seedlings for this technology to be accepted by forest growers [50]. Zygotic seedlings must reach certain specifications to be considered as optimum planting stock, the attributes considered important include such measurements as shoot height, root length, diameter at the root collar, and combinations of these. Such measurements can be a good estimate of the seedling quality and how they will perform in the field [51].

The root development of seedlings is also a critical factor for their successful establishment when out-planted [52]. Typical sorting criteria for the acceptability of radiata pine seedlings include a measurement of root spread [53]. Due to the subjectivity of these measurements, in this work, we decided to err on the side of caution, grouping root quadrant scores from all surviving somatic emblings from each of the germination/acclimatization protocols into only two groups, representing either poor root architecture (a score of 0, 1 or 2) or good architecture (a score of 3 or 4), even though scores of 2 would be acceptable in some cases by growers.

When embryos were germinated and acclimatized under Treatment 2 (cold storage/short germination/nursery), 72% of emblings fell into the “good” root architecture category, compared to only 15% of emblings from Treatment 1 (short germination/micro-plug) and 10% of emblings from the control treatment (Figure 6). This improvement could be due to emblings being planted directly into large-volume containers with no restrictions on root growth. This protocol also avoided the need for transplanting delicate emblings, where there is the likelihood of root damage due to handling [54]. Lamhamedi et al., found that the manual transplantation of *Picea glauca* somatic plantlets resulted in numerous root deformations compared to zygotic seedlings where no transplantation was required [55]. Growing seedlings beyond the desirable level for their root mass in a media plug will negatively affect the quality of the root system and subsequent performance in the field [52]. It is possible that in the protocol used here for radiata pine somatic emblings, they were left for too long in the micro-plugs, and there may be an improvement in root architecture if the time were reduced and emblings were transplanted to larger containers earlier.

Reducing the requirements for strict environmental conditions for acclimatization would greatly decrease the overall cost of the SE technology, as this is one of the challenges with integrating tissue-cultured plantlets into normal seedling production [32]. To be able to do this, encouraging emblings to move toward an autotrophic state while still in vitro is required [56]. This can be done by altering the in vitro environment, such as reducing or removing the sugar in the medium, reducing the humidity with the use of culture vessels with ventilation [21,57], or simply opening the vessels for increasing periods [58]. We believe the use of the Vivi™ micro-plug system, in combination with the shortened germination period, played a large role in the improved quality of radiata pine somatic emblings. The combination of these two factors, however, needs to be managed carefully for different species due to the small volume size of the trays. Both the root length following germination and the time emblings spend in the micro-plugs need to be optimized to prevent root abnormalities. A total of 18 cell lines were used in the radiata pine study, giving us confidence that this protocol will be advantageous for a wide range of genotypes. This is an important consideration from a commercial grower’s point of view.

In conclusion, a germination and acclimatization protocol designed for Norway spruce which includes a shortened germination period and a micro-plug system for in vitro to ex vitro transfer of somatic emblings, has successfully been applied to two pine species. In *P. radiata*, this protocol significantly improved the chances of survival as well as improving the overall quality of resulting emblings. An even more simplified protocol involving a cold storage step, shortened germination period and direct transfer to nursery conditions also showed some promise, especially with respect to root architecture in radiata pine.

For *P. sylvestris*, the current results show that shortened in vitro germination can give as good survival of the emblings as the traditionally used longer, two-step in vitro germination method. This needs to be confirmed, however, using broader material. Further, for this species, the earlier phases, especially initiation, are challenging and require improvements for SE to be practically applicable in this species.

The measurement regime of embryos and emblings applied to Scots pine could be refined and applied to other species. Root length or overall length after in vitro germination may work as an indicator for post-transplant survival, or it could be applied as a threshold to determine if embryos are qualified for transplanting and/or to determine the optimal length of the germination period. However, a larger dataset is required for this, and the measurements would need to be optimized for each growing system (container plug size) and species.

For *P. radiata*, we believe that the improvements to the protocol presented here could be optimized further. Both the time on germination and the time spent in Vivi™ trays have the potential to be reduced. Both adjustments are likely to improve the quality of the root system. It could also be advantageous to investigate replacing the loose fill in the Vivi™ trays used in this study with small miniplugs, therefore avoiding or limiting root disturbance during transplant from these trays to larger containers. Further conditions in the germination stage could also be investigated for radiata pine in our protocols, such as nitrogen content and light quality. Long-term field trials of somatic seedlings resulting from these protocols, as well as comparisons with currently acceptable planting stock, will be required to ensure the issues with quality historically seen by industry using direct SE are not still an issue. Cold storage (both conditions and timing) should be investigated further. Although some changes seen while in cold storage may be beneficial, it is important to reduce the likelihood of negative changes (e.g., precocious germination).

## 4. Materials and Methods

### 4.1. Pinus radiata

#### Plant Material

Mature somatic embryos from 18 *P. radiata* embryogenic cell lines were used for germination experiments. These cell lines were initiated at Scion, Rotorua, NZ in 2010 (4 lines, cell lines A–D) and 2021 (14 lines, cell lines E-R) using protocols described by Hargreaves [59]. Briefly, green cones from control pollinated elite families containing immature zygotic embryos were collected from a commercial seed orchard (Proseed) located at Amberly, North Canterbury, NZ. Seeds were removed, and the surface was sterilized and cultured as either dissected zygotic embryos (2010) or whole megagametophytes containing zygotic embryos (2021). The culture medium was a Litvay-based medium [60], as modified by Hargreaves et al. [61].

All cell lines, whether recovered from cryopreservation using protocols developed for *P. radiata* [62] (A–D) or newly initiated (E–R), were proliferated on a modified Litvay-based medium [60] supplemented with 4.5 µM 2,4-D, 2.2 µM 6-BAP and 30 g/L sucrose (for the full composition of proliferation medium see Table 3). The medium was solidified with 3 g/L Gelrite, and pH was adjusted to 5.6–5.8.

### 4.2. Methods

#### 4.2.1. Maturation of Somatic Embryos

Following a proliferation phase, embryogenic cell lines were then transferred as lumps (approximately 5 × 100 mg per Petri plate) to a pre-maturation medium (proliferation medium minus plant growth regulators) for two weeks. Tissue was then suspended in a proliferation liquid medium containing no growth regulators at a rate of 1 g of embryogenic tissue to 10 mL of liquid medium. Then, 1 mL of this suspension (100 mg of embryogenic tissue) was dispensed to filter paper discs (MS Grade 1, 70 mm) on a maturation medium [63] for the development of mature somatic embryos. The medium was supplemented with filter sterilized 56.75 µM ABA and 60 g/L sucrose and was solidified with 8 g/L Gelrite (for the full composition of maturation media, see Table 3). Cultures were incubated in the dark at 24 ± 1 °C for all stages. A sample of maturation plates containing mature somatic embryos (after approximately 10–12 weeks) of 10 of the cell lines (A–J) were stored in cold storage (5–6 °C) for a period of 5–6 months.

#### 4.2.2. Germination Treatments and Further Growth of Emblings

Mature cotyledonary somatic embryos were harvested from a maturation medium and placed on a modified Boulay [64] germination medium, KNV87 [65] (for the full composition of the germination medium, see Table 3). The medium was supplemented with 5 g/L activated charcoal with the addition of 30 g/L sucrose and solidified with 3.5 g/L Gelrite. Embryos were placed horizontally on the surface, and their bases were gently pressed into the medium. Embryos were incubated at 24 ± 1 °C in the dark for 7 days, followed by 50% shade cloth to partially expose the somatic embryos to a light intensity of 40 µmol m^−2^ s^−2^ for a further 7 days and then full light intensity of approximately 90 µmol m^−2^ s^−2^ for the remainder of the germination period (3 weeks).

For the somatic embryos which were not cold stored (18 cell lines), two germination treatments were compared; in the control treatment, somatic embryos were cultured on KNV87 for a total of 5 weeks. This is our standard germination treatment for radiata pine somatic embryos. Germinated embryos were then carefully removed from germination media. Roots were trimmed where necessary and transferred to the glasshouse facility (23.5 °C, 68% RH), where they were planted into seed propagation trays containing a screened bark fiber medium: perlite: vermiculite: 3 mm pumice: peat mix (40:10:10:20%). Propagation trays were covered with plastic vented lids (vent closed) and covered with 50% shade cloth for one week, and misted daily. The shade cloth was then removed, and emblings were slowly hardened using the vent and the eventual removal of the lid over 4–6 weeks.

In Treatment 1 (short germination/micro-plug protocol), somatic embryos were cultured on KNV87 for 2 weeks, as described above, before being planted into Vivi™48 (ViVi Pak, ViVi) trays containing a sterile loose mix of peat: perlite (30:70%). Trays were placed inside Vivi™ outer boxes and covered with perforated foil (15 rows). Trays were then returned to the light rig under full light conditions for a further 3 weeks. Boxes containing trays were then transferred to a propagation room in a glasshouse facility (23.5 °C, 68%RH). Three × 15–20 cm slits were made in the foil at this stage, and emblings were hand watered through these slits for a further 3 weeks. The foil was then removed completely, and emblings were then misted with a hose. A total of between 10 and 72 individual somatic embryos for each cell line were used for each of the treatments depending on the embryo availability of each cell line.

After approximately 6–7 weeks under the above conditions, all surviving emblings from both treatments were repotted into Transplant systems TS48 trays containing 30% No. 8 composted pine bark, 35% C.A.N Fines (fine-screened composted pine bark) and 35% Coco Fiber with 6 kg per cube of osmocote exact control release fertilizer, 0.75 kg per cube micronutrients. At this time, an assessment of survival was carried out. Two weeks later, the trays were moved to another room in the glasshouse with natural light conditions. A shade cloth was used for approximately 2 weeks to allow for adaptation to these new conditions. A final assessment was done on emblings approximately 13–15 weeks after repotting. The following characteristics were monitored at the final assessment; survival (%), shoot height (mm), root length (mm), root collar diameter (mm) and root quadrant score (0–4). The definitions for root quadrant scoring were 0 = primary root only (i.e., there was no further development of the root system, 1 = one quadrant of the primary root occupied with secondary/tertiary roots, 2 = two quadrants occupied with secondary/tertiary roots, 3 = three quadrants occupied with secondary/tertiary roots, 4 = four quadrants (all) occupied with secondary/tertiary roots.

For 10 of the 18 cell lines tested in the control treatment and Treatment 1, a third treatment was also tested, Treatment 2 (cold storage/short germination/nursery). In this treatment, embryos were harvested from a maturation medium following a period in cold storage (5–6 months) and placed on KNV87 as above for 2 weeks. Embryos were then transferred to the nursery, removed from the culture, and planted into Ellepot 4 cm plugs (125 ccs) containing no.8 bark media (see Table 4 for full composition) in a nursery rooting facility, with a starting relative humidity of 85% and stepping down by 5% every week to a minimum of 70%, a bench temperature of 25 °C, and 2 min of irrigation 3 times a day for 6 weeks. After 6 weeks, emblings were then transferred to a nursery facility with a relative humidity of 60%, a bench temperature of 20 °C and 2 min of irrigation once a day for 4 weeks and finally, an open growing environment with no humidity or heat control (maximum 26 °C), 30 min of watering every 3 days and 30-min fertigation (high N, low P and K) once a week. Emblings were assessed for the above growth characteristics after approximately 19 weeks in these conditions. All treatments are described by illustration in Figure 1 in the Results section.

#### 4.2.3. Experimental Design and Statistical Evaluation

Survival at final was calculated as the number of viable plants at the end of the experiment divided by the number of viable plants at repotting. As the experiment was not fully factorial, differences in shoot length, root length and root collar diameter between treatments and cell lines were tested using nested analysis of variance. Pairwise contrasts were subsequently tested using Duncan’s tests (procedure GLM of SAS).

The root quadrant score (RQS) is a measurement of how balanced the root architecture is and, therefore, how likely the plant will be able to support itself. An RQS of 0 means that only the primary root is present. Scores from 1 to 4 represent further growth of the root system, with the number representing the area of further root growth if the center of the primary root was split into 4 quadrants. A rating of 2 is unbalanced, with all further root development occurring in only two of these quadrants, and a rating of 4 is very well-balanced, with all four quadrants containing further root growth. For statistical evaluation of the root quadrant score, scores (0–4) were grouped into two categories, poor root architecture (a score of 0, 1 or 2) and good root architecture (a score of 3 or 4). For statistical evaluation of the root quadrant score, scores (0–4) were grouped into two categories, poor root architecture (a score of 0, 1 or 2) and good root architecture (a score of 3 or 4). Differences in the frequency of categories among treatments and among cell lines were tested using a log-likelihood χ2 test (procedure FREQ of SAS; SAS 2010).

### 4.3. Pinus sylvestris

#### 4.3.1. Plant Material

Cryopreserved samples of Scots pine (*P. sylvestris*) embryogenic cell lines were thawed in August 2020 following established procedures. Briefly, samples were rapidly thawed in a water bath, and the cryoprotectant was rinsed off with liquid proliferation media. Samples were then plated on proliferation media with filter paper and transferred three times, gradually decreasing the sucrose content in the proliferation media [66]. Overall, 7 cell lines were recovered from cryopreservation and proliferated (subcultured bi-weekly) on MB3 medium consisting of a DCR-based maintenance medium (DCR3), containing 9.1 µM 2,4-D and 2.2 µM BA and solidified with 2.5 gL^−1^ Phytagel [12,67,68].

#### 4.3.2. Methods

##### Production of Somatic Embryos and Cold Storage

Maturation was carried out in October 2020 for seven cell lines and repeated in November 2020 by weighing and suspending approximately 150 mg of embryogenic tissue in liquid MB4 medium (with 10 g/L active charcoal, without hormones) which was then dispensed onto filter paper (Whatman #2; 55 mm diameter) using a Büchner funnel as described by Klimaszewska and Smith [20]. Maturation in both trials was carried out on MB4 maturation medium containing 80 µM ABA, 500 mg/L casein hydrolysate, 250 mg/L L-glutamine, and 200 mM sucrose and 9 gL^−1^ Phytagel, modified from [12] for a period of 10 weeks in the dark (+24 °C). Maturation plates containing mature somatic embryos of each cell line were stored in cold storage (+2 °C) [11] for a period of 6–7 months prior to germination. A total of 870 embryos from seven cell lines (matured in November) were cool stored and germinated in three treatments in June 2021 (June21 trial).

##### Germination Treatments and Further Growth

In December 2020 (DEC20 trial), 216 cotyledonary embryos from four of the cell lines (54 embryos per cell line) were germinated in vitro directly following maturation carried out in October.

In vitro, germination for both trials was carried out in a +20 °C growth room [11,16]. LEDs (Valoya AP67) were used to provide lighting for the germinating embryos at light intensities of 5, 50 and 150 µmol/m^−2^/s^−1^ in time proportions of 43.0, 28.5, and 28.5%, respectively.

Three different germination treatments were studied in Scots pine. Control treatment consisted of germination on MB5 Petri plates for two weeks with increasing light intensity as described above, and two weeks in MB6 jars in “full light” following the established Scots pine procedure [12]. In MB 2 w treatment, cotyledonary embryos were germinated on MB5 Petri plates for two weeks with increasing light intensity. Control and MB 2 w treatments were studied in both trials: Dec20 (four cell lines, directly after maturation) and June21 (cold stored from both Oct and Nov maturations, seven cell lines). In the June21 trial, a third treatment was introduced (LM 2 w, cold stored embryos only), in which cotyledonary embryos were germinated for two weeks in Petri plates on LM media modified for Norway spruce (*Picea abies* L. Karst) in increasing light intensity following the procedure successfully used for Norway spruce [11,16].

Following germination treatments, emblings were transplanted to a peat-based growing substrate. In the Dec20 trial, the transplanting was carried out to miniplugs (plug volume 3.4 mL, +Preforma 126/JIF, ViVi Pak, ViVi, Burgh Haamstede, The Netherlands) and the emblings were grown for ten weeks in the growth room, according to the protocol by Tikkinen et al. [11]. The final inventory was carried out ten weeks after transplanting. In the June21 trial, the emblings were transplanted to Plantek Pl 81 f containers (81 seedlings per tray, volume: 85 cm^3^, cell surface area: 18.3 cm^2^, growing density: 546 cells m^−2^; BCC, Iso-Vimma, Finland) filled with semi coarse, pre-fertilized, light peat substrate (Kekkilä FPM 420 W F6) and placed into a commercial greenhouse unit where the sowing of a seed lot had just been finished [11]. The final inventory for the June21 trial was made after the first growing season (GS1) in early October 2021. All treatments are described by illustration in Figure 8 in the Section 2.

In both trials, all emblings, independent of their quality or germination vigor, were transplanted to containers to enable tracking of the development of an individual embryo within a treatment, cell line and container location. Partially due to this, the proportion of viable plants after the growing season appears low and does not fully represent the conversion rate of germinated embryos.

##### Experimental Design and Statistical Evaluation

Cotyledonary embryos were photographed (Canon Powershot G5 PC1049) in Petri plates containing a scale at the beginning of and after two weeks of in vitro germination. Additionally, the emblings were measured after one week of germination in the Dec20 trial to determine the growth rate of the emblings during the first and second weeks of germination. At the beginning of in vitro germination, the overall length of the cotyledonary embryo was measured. After one and two weeks, the length of the shoot, root and overall length were measured. All the measurements were carried out in pixels (ImageJ software (version 1.53 c; Java 1.8.0_312 [64-bit]), which were then converted to millimeters by applying the scale in individual photos.

To compare survival of emblings in different size classes the embling were grouped into following categories: 1 mm classes for shoot length (0 ≤ x < 1; 1 ≤ x < 2; 2 ≤ x < 3; 3 ≤ x < 4; 4 ≤ x < 5; 5 ≤ x < 6; 6 ≤ x < 7; 7 ≤ x < 8; 8 ≤ x < 9; 9 ≤ x < 10; 10 ≤ x < 11 mm), 2 mm classes for root length (0 ≤ x < 2; 2 ≤ x < 4; 4 ≤ x < 6; 6 ≤ x < 8; 8 ≤ x < 10; 10 ≤ x < 12; 12 ≤ x < 14; 14 ≤ x < 16; 16 ≤ x < 18; 18 ≤ x < 20; 20 ≤ x < 22; 22 ≤ x < 24; 26 ≤ x < 28 mm) and 3 mm classes for total length (0 ≤ x < 3; 3 ≤ x < 6; 6 ≤ x < 9; 9 ≤ x < 12; 12 ≤ x < 15; 15 ≤ x < 18; 18 ≤ x < 21; 21 ≤ x < 24; 24 ≤ x < 27; 27 ≤ x < 30; 30 ≤ x < 33 mm).

All measurements and inventory results were analyzed using SPSS Statistics version 27. Normality could not be assumed among embryo and embling measurements. Thus, the Mann–Whitney U nonparametric test was applied to compare the size of embryos and emblings between trials, cell lines and treatments. The Kruskall–Wallis nonparametric test was used to compare survival differences among different embling size classes. When the December 20 and June 21 trials were compared, the analysis was made for only the cell lines included in both trials.

## Figures and Tables

**Figure 1 plants-12-02095-f001:**
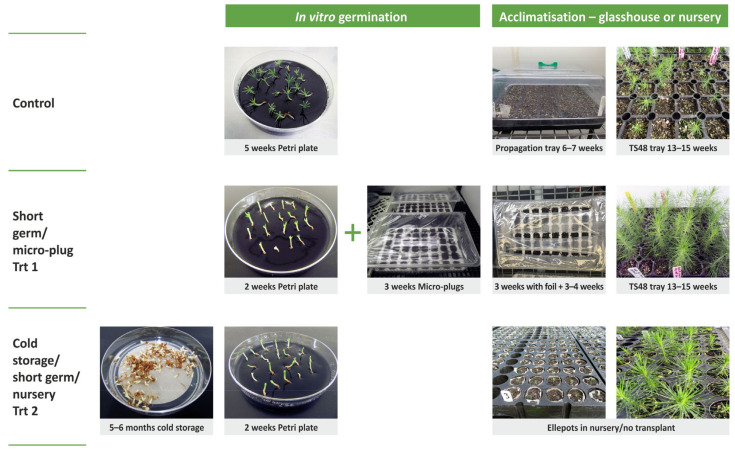
Experimental design of variants (treatments) tested with somatic embryos of P. *radiata*. Three treatments; Control (germination on Petri plates for 5 weeks, then to propagation trays in the glasshouse for six to seven weeks before being transplanted), Trt 1 (germination on Petri plates for two weeks, then to a micro-plug system for three weeks in vitro followed by transfer to a glasshouse for a further six to seven weeks before being transplanted, and Trt 2 (five to six months of cold storage followed by germination on Petri plates for two weeks before being transplanted directly to a nursery).

**Figure 2 plants-12-02095-f002:**
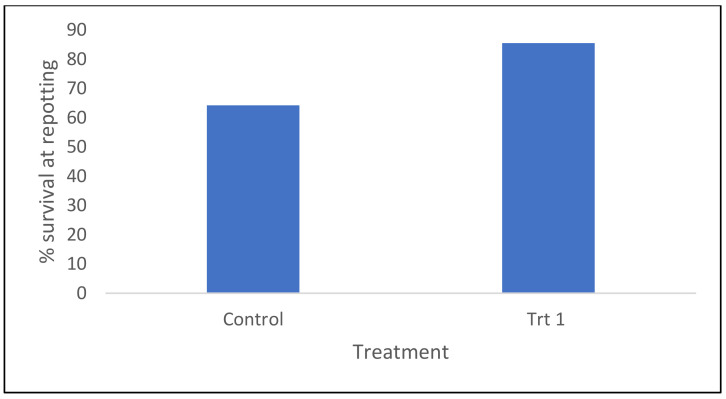
Survival rates of *P. radiata* emblings at the repotting stage for the Control and Trt 1 (short germination/micro-plug protocol), all cell line data pooled.

**Figure 3 plants-12-02095-f003:**
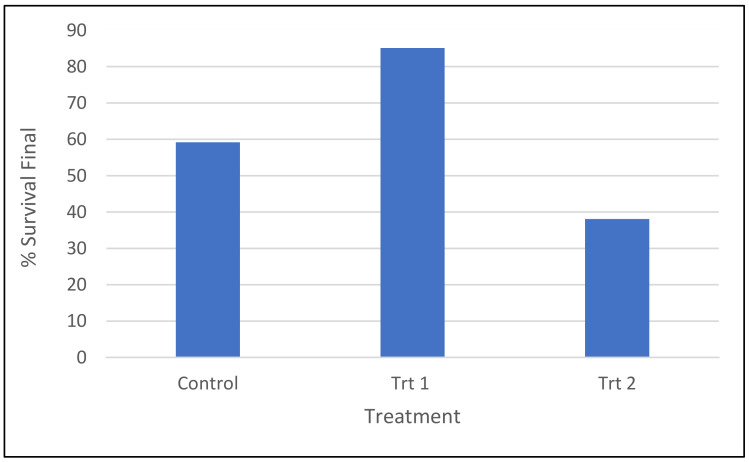
Survival percentages of *P. radiata* emblings in the three treatments: Control, Trt 1 (short germination/micro-plug protocol) and Trt 2 (cold storage/short germination protocol) by the end of the experiment, cell line data pooled. For control and Trt 1, these are the percentage of emblings surviving following repotting.

**Figure 4 plants-12-02095-f004:**
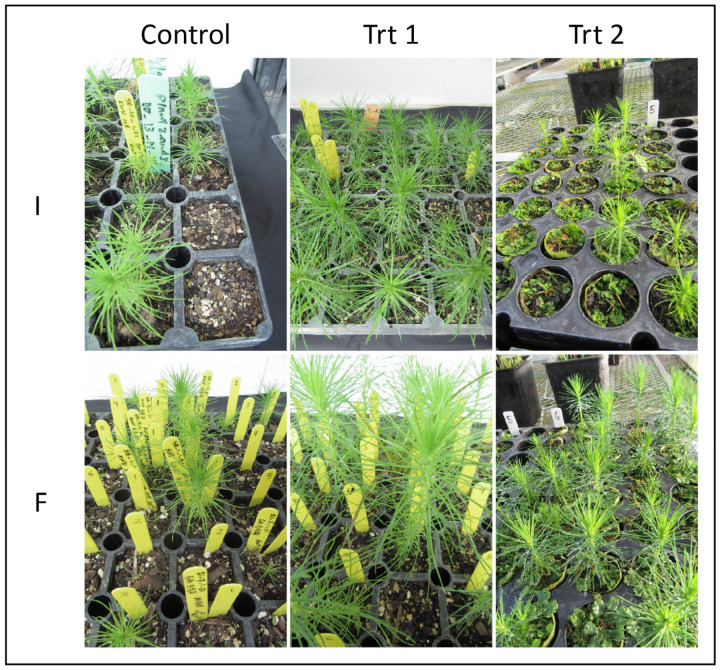
Somatic emblings of *P. radiata* originated from two cell lines (I and F) being exposed to the three germination/acclimatization protocols (Control, Trt 1—short germination/micro-plug protocol and Trt 2—cold storage/short germination protocol) by the end of the experiment.

**Figure 5 plants-12-02095-f005:**
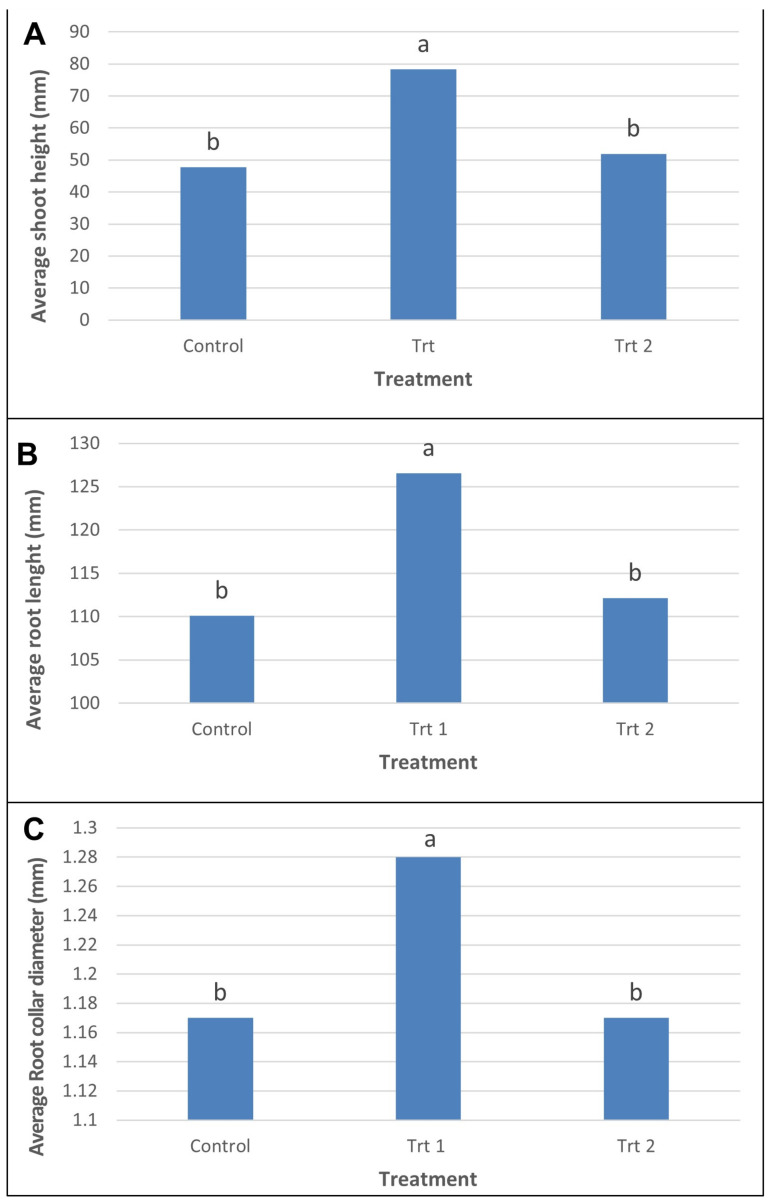
Statistical evaluation of pairwise contrasts for the mean shoot height (**A**), root length (**B**) and root collar diameter (**C**) of *P. radiata* emblings observed during tested germination treatments: control, short germination/micro-plug protocol (Trt 1), and cold storage/short germination protocol (Trt 2). Bars with different letters denote significant (*p* < 0.05) differences.

**Figure 6 plants-12-02095-f006:**
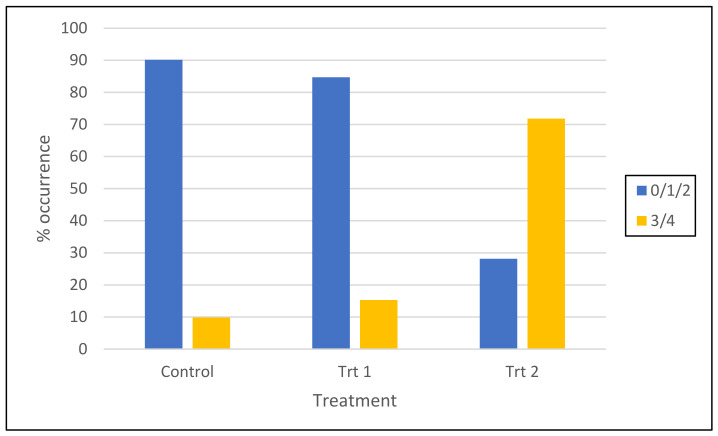
Comparison of the effects of three tested germination treatments (control, Trt 1 (short germination/micro-plug protocol) and Trt 2 (cold storage/short germination protocol)) on the root architecture of *P. radiata* emblings presented as root quadrant score categories (cell line data pooled).

**Figure 7 plants-12-02095-f007:**
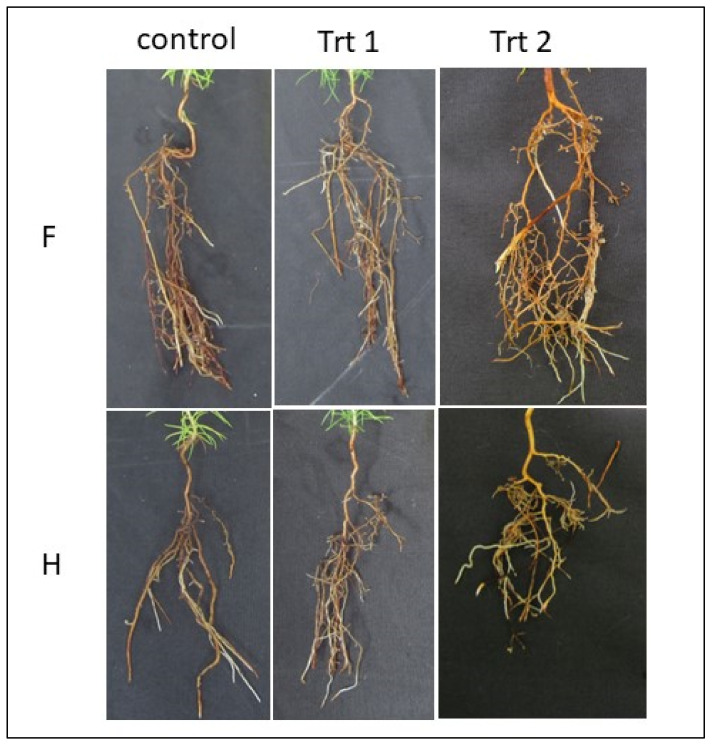
Root structures of somatic emblings of *P. radiata* originating from two cell lines (F and H) being exposed to the three germination/acclimatization protocols (Control, Trt 1 (short germination/micro-plug protocol) and Trt 2 (cold storage/short germination protocol) by the end of the experiment).

**Figure 8 plants-12-02095-f008:**
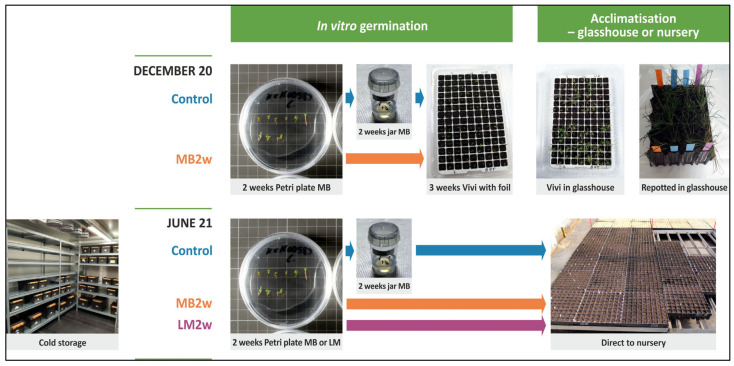
Experimental design of variants (treatments) tested with somatic embryos of *P. sylvestris*. Three treatments; Control (germination on MB5 Petri plates for two weeks + two weeks in MB6 jars), MB 2 w (germination on MB5 Petri plates for two weeks) and LM 2 w (germination on LM Petri plates on LM media modified for Norway spruce (*Picea abies L.Karst*) for two weeks) in two trials; December 2020 (DEC20) and June 2021 (June21).

**Figure 9 plants-12-02095-f009:**
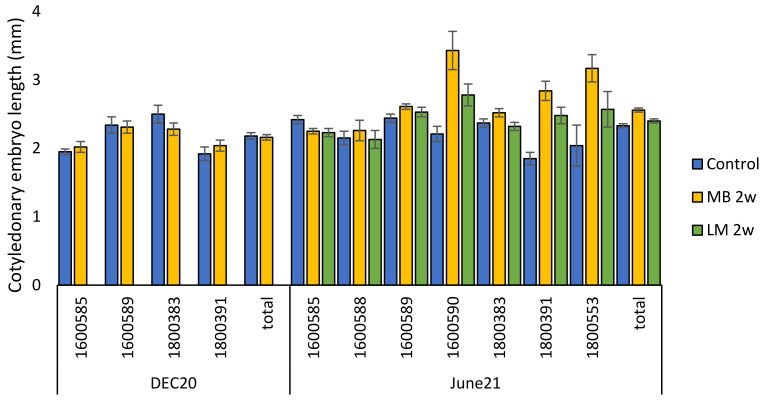
Length of *P. sylvestris* cotyledonary embryos (mm) before in vitro germination in different cell lines and overall, in treatments Control, MB 2w and LM 2w, in the December 2020 (DEC20) and June 2021 (June21) trials. Mean values ± SEM (standard error of the mean).

**Figure 10 plants-12-02095-f010:**
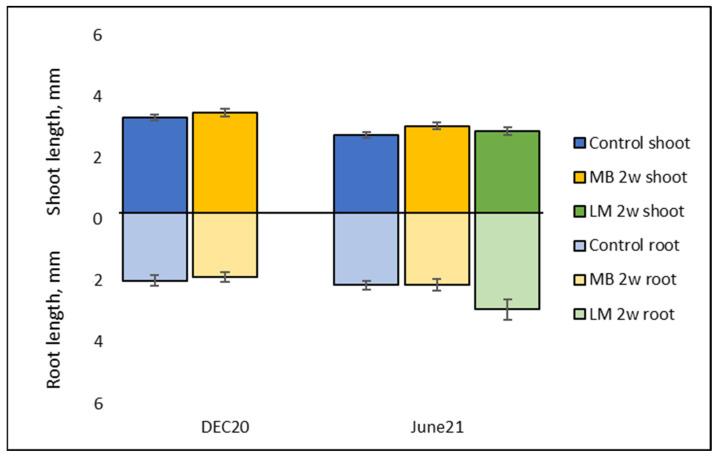
Length of *P. sylvestris* emblings in treatments Control, MB 2 w and LM 2 w after two weeks of in vitro germination (shoot in positive axis, root in negative axis) in DEC20 and June21 trials, all cell lines pooled. Mean values ± SEM (standard error of the mean).

**Figure 11 plants-12-02095-f011:**
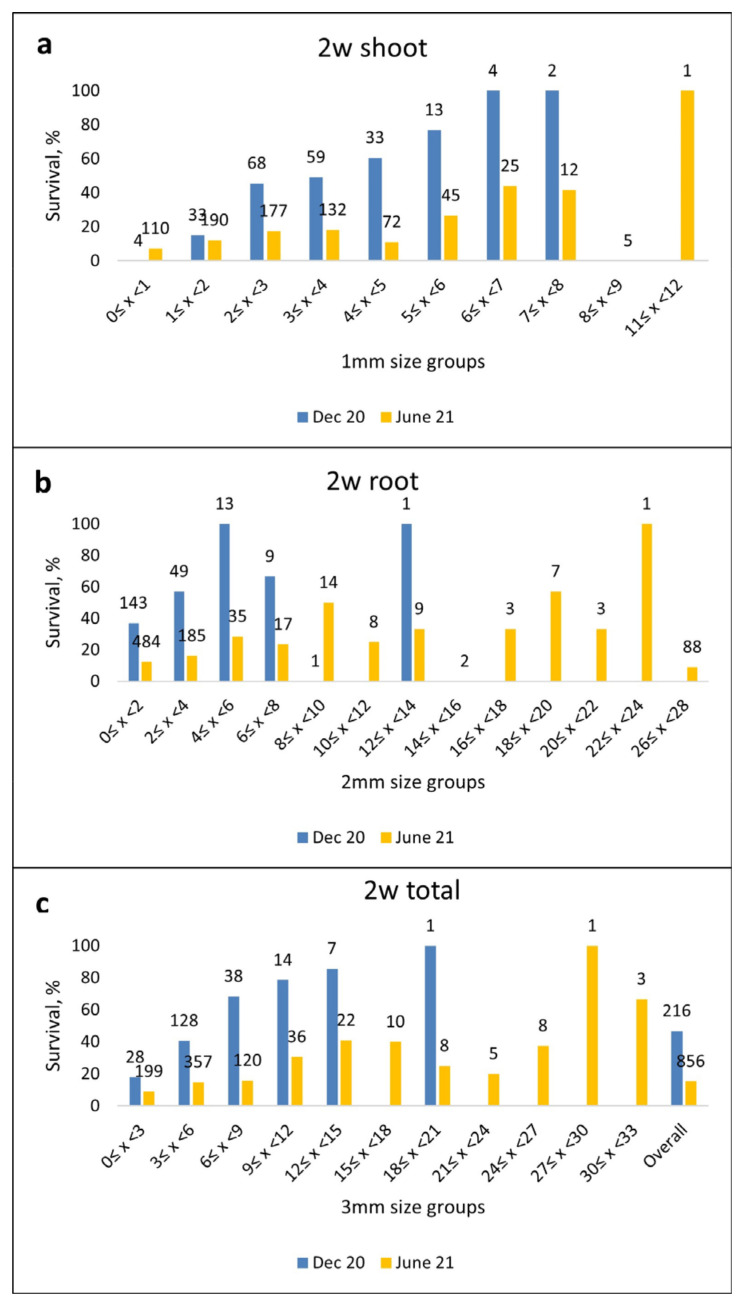
Mean survival (%) in different size classes (**a**) shoot (1 mm classes), (**b**) root (2 mm classes) and (**c**) total length (3 mm classes) of *P. sylvestris* emblings and overall, after in vitro germination of two weeks on Petri plates in both trials (DEC20 and June21). The number of samples is presented on data labels. The figures include bars from all size classes where at least one embling was found.

**Table 1 plants-12-02095-t001:** Analysis of variance of the effects of germination treatments (control, short germination/micro-plug (Trt 1), cold treatment/short germination (Trt 2)) across tested embryogenic cell lines of *P. radiata* on the shoot and root heights and root collar diameter. Significance label: *** *p* < 0.001.

Source	Shoot Height		Root Length		Root Collar Diameter	
	DF	*F* Value	DF	*F* Value	DF	*F* Value
Treatment	2	153.90 ***	2	38.31 ***	2	25.77 ***
Cell line (treatment)	43	9.23 ***	43	6.22 ***	43	8.58 ***
Error	731		731		731	

**Table 2 plants-12-02095-t002:** Proportion of *P. sylvestris* germinated embryos in the two trials (DEC20 and June21) and three treatments; Control, MB 2 w and LM 2 w (when length thresholds of 3 mm overall and 2 mm roots were applied was considered as a germinated embryo), shoot length, survival % (of all embryos placed to germination) after first growing season (GS1), root length and total length of emblings after two weeks of in vitro germination. Mean values are presented with the standard error of the mean.

Trial	DEC20			June21			
Treatment	Control	MB 2 w	Total	Control	MB 2 w	LM 2 w	Total
Germination, %	37.0	29.6	33.3	41.1	29.5	28.5	32.9
GS1 Survival, %	48.1	45.4	46.8	28.6	7.0	10.2	15.3
2 w shoot, mm	3.83 (±0.18)	4.20 (±0.27)	3.99 (±0.16)	3.82 (±0.15)	4.47 (±0.19)	3.84 (±0.19)	4.03 (±0.1)
2 w root, mm	3.80 (±0.36)	3.92 (±0.33)	3.85 (±0.24)	4.18 (±0.27)	5.68 (±0.54)	6.51 (±0.7)	5.28 (±0.28)
2 w total, mm	7.64 (±0.49)	8.12 (±0.51)	7.85 (±0.35)	8.00 (±0.37)	10.15 (±0.66)	10.35 (±0.81)	9.31 (±0.34)

**Table 3 plants-12-02095-t003:** Composition of media used for the proliferation, maturation, and germination of *P. radiata* embryogenic cell lines.

Component Group	Chemical	Modified Version [61] of Litvay Medium [60] Used for Proliferation Stage	Modified Version [61] of Litvay Medium [60] Used for Maturation Stage	Modified Version [63] of Boulay Medium [64] Used for Germination
		[mg/L]	[mg/L]	[mg/L]
Macronutrient	KH_2_PO_4_	170	170	272
	KNO_3_	950	950	505.5
	NH_4_NO_3_	825	825	360
	MgSO_4_·7H_2_O	925	925	493
	CaCl_2_·2H_2_O	11	11	
	Ca(NO_3_)_2_·4H_2_O			708.5
	KCl			149
Micronutrient	MnSO_4_·H_2_O	21	21	4.23
	H_3_BO_3_	31	31	4.64
	ZnSO_4_·7H_2_O	43	43	1.44
	KI	4.15	4.15	0.083
	Na_2_MoO_4_.2H_2_O	1.25	1.25	0.12
	CuSO_4_·5H_2_O	0.5	0.5	0.25
	CoCl_2_·6H_2_O	1.25	1.25	0.012
	AlCl_3_·6H_2_O			0.024
	NiCl_2_·6H_2_O			0.024
Chelated iron	FeSO_4_·7H_2_O	30	30	13.9
	Na_2_EDTA·2H_2_O	40	40	18.625
Vitamin	Nicotinic acid	5	5	0.5
	Pyridoxine-HCl	0.5	0.5	0.5
	Thiamine	5	5	1
	myo-inositol	100	100	100
Amino acids	glycine	2.0		2.0
	glutamine	450	550	
	asparagine		525	
	arginine		175	
	citrulline		19.75	
	ornithine		19	
	lysine		13.75	
	alanine		10	
	proline		8.75	

**Table 4 plants-12-02095-t004:** Composition of no. 8 bark media used in Ellepots for somatic emblings growing on in nursery rooting facility in Treatment 2.

Component	Media Type	Amount
Besgro Precision No. 8 Bark (3–6 mm)	Composted pine bark	30%
C.A.N Fines fine screened	Composted pine bark	35%
Coco Fibre precision	Coir	35%
Osmocote Exact 12/14 Protect DCT	Control release	6 kg/m^3^
Microplus (TE + Mg + Fe)	Micronutrients	0.75 kg/m^3^
Dolomite	Calcium magnesium carbonate	1 kg/m^3^
Gypsum fine	Calcium sulfate dihydrate	2 kg/m^3^
Hydraflo 2 Granular Wetting Agent	Wetting agent	0.3 kg/m^3^

## Data Availability

Radiata pine data will be made available upon request from Cathie Reeves, and Scots pine data will be made available upon request from Mikko Tikkinen.

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
