# Peer review of "Application of Cold Storage and Short In Vitro Germination for Somatic Embryos of Pinus radiata and P. sylvestris"

_plants, 2023, doi:10.3390/plants12112095_

Round 1

Reviewer 1 Report

The need for resilient, well-acclimated and hardy emblings derived from SEs is well-known to those in the conifer SE field, and this manuscript provides a good contribution to the Pinus literature regarding this aspect. I enjoyed reading this paper, and I think that with some modifications, it will be an informative addition to the pine/conifer knowledge base.

1. I found myself flipping back and forth between the Results and the Materials and Methods sections to have a good understanding of the various treatment details as I was assessing the Results. One thing I think that would help the reader would be to have a Table at the start of the Results section, describing the Control and Treatment conditions. This would be handy as the reader dives into the Results. Perhaps a Table coupled or integrated with Schemes 1/II would work here?

2. In Section 2.1, the authors make a point of mentioning that materials from Lines A-D were from cryopreserved stocks. However, there is no further mention of the cryopreserved materials relative to the other newly induced lines. Did the authors check to see if these lines exhibited any differences from non-cryopreserved materials? If not, it might be worth checking and commenting. I had a quick look at the numbers, and it seemed to me that the average values for the combined A-D lines were pretty much lower than the averages for the combined E-J lines in both Tables S1 and S2.  The trends appear to be the same, with Treatment 1> Control and Treatment 2, but the combined average values for A-D are lower than combined E-J. If this is indeed the case, does this indicate that the previously cryopreserved lines don’t respond as well as the freshly induced lines, or that these 4 genotypes are just generally poorer?

3. Line 221 - Shouldn't this be treatment 1?

4. Section 2.2.1 - As soon as you view Figure 7 and Figure, you encounter MB and LM components, however there is no mention of these treatments until much later. This is another reason why I think having a description of the treatments (as mentioned earlier) in the results section would be beneficial to the reader.

5. Figure 7 and 8 Legends - Mean values + SEM

6. You have Scheme 1, but Scheme II. Use a consistent format i.e., Scheme I and Scheme II OR Scheme 1 and Scheme 2.

7. I have a problem with the Y-axis labels for Figures 8, S1 and S2. To me, the -2, -4 and -6 numbers denote negative growth. Change these to positive numbers, and on the Y-axis, label the Y-axis above 0 as Shoot Growth, and the Y-axis below 0 as Root Growth. Also, list the value units (mm).

Reviewer 2 Report

The article entitled "Application of cold storage and short in vitro germination for somatic embryos of Pinus radiata and P. sylvestris" by Reeves and colleagues presents a valuable contribution to the field of plant biotechnology for these species. The authors provide insightful results on the effects of cold storage and improved protocol for somatic embryos of Pinus radiata and P. sylvestris. However, the article is quite heavy in information, and the results section is dense and difficult to follow, specially due to lack of information in this section. Therefore, it is necessary to bring more clarity to the article. Despite this, the paper provides relevant information that brings scientific value. To improve the article, the title needs to be improved, the figures and tables should be self-explanatory, and the authors should present germination/conversion data for all treatments. In this way, the article can be more accessible to the reader and achieve its full potential.

The title needs improvement, as "short in vitro germination" does not sound good. A better title could be "Cold storage of Pinus radiata and P. sylvestris somatic embryos: towards development of an efficient protocol".

All figures and tables should be self-explanatory, especially in this journal, where the materials and methods section comes after the results. In the current format, the reader is forced to go back and forth between sections to understand the treatments and evaluation.

As a suggestion, in the first paragraph of the results, the authors should cite schematic examples of the treatments, explaining all treatments succinctly. This will allow the reader to better follow the text and results.

The percentage of germination among treatments is not presented, only the survival rate of seedlings. The germination/conversion data will be very important. In the current analysis, the results may have been induced to be relevant, but, for example, if 100 embryos germinated in the control group, but "only 64%" survived, it still represents 64 seedlings. However, in the first treatment, only 10 embryos germinated, and 86% survived, which would produce "only" 8.6 seedlings.

Round 2

Reviewer 2 Report

The present manuscript underwent significant changes to enhance readability and comprehension, and the authors had their point of view regarding other points, which is acceptable. As a result, the article can be published in its current form.